# Infrared Sensation-Based Salient Targets Enhancement Methods in Low-Visibility Scenes

**DOI:** 10.3390/s22155835

**Published:** 2022-08-04

**Authors:** Hongjun Tan, Dongxiu Ou, Lei Zhang, Guochen Shen, Xinghua Li, Yuqing Ji

**Affiliations:** School of Transportation Engineering, Tongji University, No. 4800 Caoan Road, Shanghai 201804, China

**Keywords:** infrared image, wavelet transform, multi-scale retinex algorithm, salient region, deep learning, target detection

## Abstract

Thermal imaging is an important technology in low-visibility environments, and due to the blurred edges and low contrast of infrared images, enhancement processing is of vital importance. However, to some extent, the existing enhancement algorithms based on pixel-level information ignore the salient feature of targets, the temperature which effectively separates the targets by their color. Therefore, based on the temperature and pixel features of infrared images, first, a threshold denoising model based on wavelet transformation with bilateral filtering (WTBF) was proposed. Second, our group proposed a salient components enhancement method based on a multi-scale retinex algorithm combined with frequency-tuned salient region extraction (MSRFT). Third, the image contrast and noise distribution were improved by using salient features of orientation, color, and illuminance of night or snow targets. Finally, the accuracy of the bounding box of enhanced images was tested by the pre-trained and improved object detector. The results show that the improved method can reach an accuracy of 90% of snow targets, and the average precision of car and people categories improved in four low-visibility scenes, which demonstrates the high accuracy and adaptability of the proposed methods of great significance for target detection, trajectory tracking, and danger warning of automobile driving.

## 1. Introduction

Driving safety always has the power to fuel wide discussion among society and automatic industries. The state-of-art sensor kit of automobiles which involves a camera, lidar, radar, global positioning system (GPS), and sonar is instrumental in the current autonomous driving epoch. To enable automobiles to drive safely all day and in all weather is challenging [1]. One of the most critical issues in the development of autonomous driving sensor technologies is their poor performance under adverse weather conditions [2,3], such as nighttime, rain, snow, and fog, which is called low-visibility scenes in this paper.

Retrospective past decades, the grave problems of safety and security in adverse conditions have drawn the attention of society, and numerous studies have been performed, which have exposed the vulnerability of the functioning of transportation services in adverse conditions [4]. For example, rainy and foggy conditions can cause significant attenuation to the functions of the camera and lidar [5]. In snowy conditions, cool temperature affects a camera system because of optical and mechanical disruptions.

Thermal infrared technology (or IR technology) can overcome poor portability and difficulties in detecting objects in adverse conditions. Moreover, the far-infrared camera (8–15 μm wavelength) filters out most blooming (from headlights and other light sources) and shadows (as long as the shadow does not linger) by adding different polarizers to the lens, which allows IR radiation to pass through fog and dust particles better than the visible light [6]. Meanwhile, IR technology is widely used in military, cultural heritage, industrial, medical, biological, and civilian applications [7,8,9,10,11,12].

In low-visibility environments, the thermal reflection of pedestrians and vehicles in infrared cameras can easily distinguish living targets from the ambient background, as is shown in Figure 1, which is a prominent advantage of infrared technology over other sensors [13]. However, as far as low contrast and universe noise of infrared images, the same image enhancement method unanimously performs badly in different low-visibility environments. The pixel-level information and temperature distribution should be considered gravely as the input of the image, which requires image enhancement and edge enhancement for the first step of infrared image processing. For easier analysis of image features, images are converted to grayscale in this paper.

The biggest contribution of work described in this paper proposes three algorithms and effectively enhances infrared images based on the temperature distribution characteristics of infrared image pixel points in four different low-visibility environments. Moreover, it improves the low signal-to-noise ratio of infrared images and the accuracy of objective detection. The infrared image datasets from different environments and devices are collected, including the open-source datasets and the data collected in this paper, which are called low-visibility infrared (LVIR) datasets. The remainder of this article is organized as follows: Section 2 reviews the related studies. Section 3 describes the steps of the proposed enhancement methods and the infrared target detection method. Section 4 presents and analyzes the experimental results. Finally, Section 5 summarizes the findings of the study and explores directions for future work.

## 2. The Related Work

In recent years, the infrared camera has been a promising resolution to relieve the impact of rainfall and snowflakes. For visual cameras, an unshielded camera can easily be damaged by ice. Raindrops and snowflakes can be filtered out by pixel-oriented evaluation [14], but there is a large extent resulting in higher false detection rates, poor real-time performance, and inaccurate positioning data [15]. Heavy fog also reduces the recognition and accuracy rate of existing traffic sensors [3]. Moreover, these further lead to wrong judgments and operation errors in unmanned vehicles, causing significant risks.

In the perception of the environment, camera and radar sensors have illustrated their benefit in the visible spectrum domain, in adverse weather conditions, changing luminance, and dim backgrounds [16]. The dim target detection in an infrared image with complex background and low signal-to-noise ratio (SNR) is a significant yet difficult task in the infrared target tracking system. Technologies to deal with this issue can be classified into two main categories: restraining the background and enhancing the targets.

To restrain the background, Dong et al. [17] combined three mechanisms of the Human Visual System (HVS) [18,19,20]. Difference of Gaussians (DOG) filters which simulate the contrast mechanism were used to filter the input image and compute the saliency map which then simulated visual attention through a Gaussian window to further enhance the dim small target. Li et al. [20] adopted an improved top-hat operator to process the original image which can restrain the background well, and then processed the enhanced image with a spectral residual method. A classification preprocessing strategy system [18] was also adopted to remove noise, suppress the background, and improve the target signal-to-noise ratio. Figure 2 shows the diagram of proposed image enhancement algorithm.To enhance the dim targets, Li et al. [21] proposed a cascade method to resolve the low SNR issue in the long-range distance detection tasks of infrared dim target detection, which takes advantage of the movement continuity of targets. Zhang et al. [22] proposed a novel enhancement algorithm based on the neutrosophic sets to enhance the contrast of infrared images. In other fields, the image enhancement and reconstruction methods are also applied in active dynamic thermography (ADT) to visualize the superficial blood vessel with high contrast [23]. In practice, these small dim targets often lack detailed information or have low signal-to-noise ratios. IR images in different harsh environments have different characteristics, and using the same enhancement method applied to these images may not achieve the desired results.

For target detection and diagnosis, Fidali et al. [24] proposed a fusing method of global machine condition assessment for infrared condition monitoring and diagnostics systems. Deep learning technology has been used in the field of visible light image segmentation as well as far-infrared images [25]. For object detection in the nighttime, features of visible images become invalid and deep learning based on traffic vehicle detection is used to fuse data obtained from multi-sensors [26]. The majority of methods-based neural networks are only using visual sensors to improve the accuracy of target detection, but they generally do not function well in harsh conditions [27]. Plenty of convolutional neural network (CNN) detectors are applied and optimized for object detection in the thermal domain. Researchers have benchmarked the performance of You Only Look Once (YOLOv3) [25] with Faster-RCNN [28], Single Shot Multi-Box Detector (SSD) [29], and Cascade R-CNN [30], or augmenting the thermal image frame with the corresponding saliency maps to provide an attention mechanism [31] for pedestrian detection in the thermal domain using Faster-RCNN. 

Except for deep neural networks, classical image processing approaches are also applied for thermal object detection. Miethig et al. [4] applied the histogram of oriented gradients (HOG) and local binary pattern for feature extraction and trained the support vector machine (SVM) classifiers for object detection in the thermal domain. Currently, multi-scale transferring learning is becoming more and more outperformed. Munir et al. [32] proposed Self Supervised Thermal Network (SSTN) to explore thermal object detection to model a view-invariant model representation by employing the self-supervised contrastive multi-scale encoder-decoder transformer network.

However, detectors scarcely perform well in low-visibility conditions. Considering that the poor infrared images may cause the lack of small targets and make the detection accuracy decrease, it is necessary to explore the detection performance of the detection algorithm under low visibility after infrared target enhancement.

## 3. Proposed Methods

### 3.1. Threshold Denoising Based on WTBF

#### 3.1.1. Wavelet Coefficients Analysis

In rainy weather, the target pedestrian occupies fewer pixels, and the high-frequency part such as noise mainly comes from the ground and umbrellas. In the rainy condition, the wavelet transformation (WT) enhancement has a better enhancement effect on the human target. The core of wavelet image enhancement is to decompose the image signal into different sub-bands after using a two-dimensional wavelet transformation on the image and to enhance or reduce the noise of the wavelet coefficients of each band.

In infrared images, visual targets are mostly present in the low-frequency part, while noise and details are more present in the high-frequency part. In the beginning, a bilateral filter is used to reduce the environmental noises of IR images. Then the threshold denoising is adapted for high-frequency coefficients and nonlinear enhancement of low-frequency coefficients. Finally, the enhanced image is reconstructed by a two-dimensional inverse wavelet transform. The structure of wavelet transformation algorithm based on a bilateral filter (WTBF) is shown in Figure 3.

#### 3.1.2. Threshold Denoising

The thresholding method can use hard and soft thresholding methods to perform nonlinear enhancement of wavelet coefficients in a certain range to effectively suppress noise while performing image enhancement, mainly with hard and soft thresholding enhancement functions. The hard thresholding function is defined as follows:(1)$$W^{\prime }\left(x,y\right)=\{\begin{array}{ll} W(x,y), & \left|W(x,y)\right|\geq \lambda \\ 0, & \left|W(x,y)\right|<\lambda \end{array}$$

Soft thresholding function:(2)$$W^{\prime }\left(x,y\right)=\{\begin{array}{ll} {}[sgn(W\left(x,y\right))*(\left|W\left(x,y\right)\right|-\lambda )], & \left|W\left(x,y\right)\right|\geq \lambda \\ 0, & \left|W\left(x,y\right)\right|<\lambda \end{array}$$
where *W*(*x*,*y*) denotes the coefficients of the image after wavelet decomposition, *W*′(*x*,*y*) denotes the output coefficients after threshold filtering, *λ* is the selected threshold value, *N* is the number of wavelet coefficients on the corresponding scale, and *σ* is the standard deviation of the additional noise signal, and the selection formula is:(3)$$\lambda =\sigma \sqrt{2ln\;N}$$

### 3.2. Salient Components Enhancement Based on MSRFT

#### 3.2.1. Multi-Scale Retinex Algorithm

The retinex algorithm is a part of the wavelet transform enhancement methods, which are widely utilized and improved. When the temperature of buildings and streets is close to living targets, it can cause false detections. The retinex algorithm is to remove the illumination component related to environmental factors from the image and obtain the reflection component that can reflect the essential characteristics of the object, by processing the RGB channels separately and then synthesizing the final enhanced image. For infrared images, it can be treated as a single-channel grayscale image, so this algorithm is also effective for infrared images.

Single Scale Retinex (SSR) is the basic retinex algorithm. A given image can be decomposed into the product of the reflection component *Reflection* and the light component *Light* in the basis of the imaging process of the object.
(4)I(x,y)=Light(x,y) × Reflection(x,y)

Compared with the SSR, the multi-scale retinex algorithm (MSR) algorithm can effectively avoid the limitation of the overall effect of the image and the detailed information due to the multiple scale components used to process the image, and then the result is obtained by weighted average. Furthermore, to improve the enhancement effect of the image. The mathematical expression of MSR is:(5)$$I^{\mathrm{MSR}}(x,y)=\sum_{s-1}^{s}W_{s}I_{s}^{\mathrm{SSR}}\left(x,y\right)$$
where the *I^MSR^*(*x*,*y*) stands for MSR processing image, *s* is the scale range, *W_s_* is the weight of each scale, and $I_{s}^{SSR}\left(x,y\right)$ is the SSR processing image of *s*.

#### 3.2.2. Light Filtering and Reflection Smoothing

The halo effects appear in areas where the brightness distribution changes drastically after the original retinex algorithm is used [33]. To reduce the halo effects of light component *Light*, a low-pass filter operator, the bilateral filter (BF) can maintain edge maintenance robustness, which utilizes the spatial similarity and gray value similarity between the current point and neighboring pixels, to overcome the halo phenomenon to a certain extent. Bilateral filtering estimates the brightness of the image:(6)$$\mathrm{Light}(x,y)=k^{-1}\left(x\right)\sum_{\left(m,n\right)\in w}d(x,y,m,n)\;\lambda (x,y,m,n)\;f(x,y,m,n)$$
(7)$$k(x)=\sum_{\left(m,n\right)\in w}d(x,y,m,n)\;\lambda (x,y,m,n)$$
(8)$${d(x,y,m,n)=e}^{-\frac{1}{2}(\frac{m^{2}+n^{2}}{\sigma_{d}})}$$
(9)$$\lambda \left(x,y,m,n\right)=\{\begin{array}{ll} e^{-\frac{1}{2}{\left(k\times f(x,y)\right)}^{2}}, & \left|f\left(x,y\right)-f\left(x,y,m,n\right)\right|<k\times f(x,y)\\ e^{-\frac{1}{2}{(|f\left(x,y\right)-f\left(x,y,m,n\right)|\times \sigma_{r})}^{2}}, & \mathrm{otherwise} \end{array}$$
where *w* is the window size, $\sigma_{d}$ is the distance scale, *f*(*x*,*y*,*m*,*n*) is the value of the pixel in the area centered at the current point (*x*,*y*), *k* is the threshold, and $\sigma_{r}$ is the illuminance scale.

For the problem of the dark tone of reflection component *Reflection*, the Gamma function is commonly used to smoothly expand the light and shade of an image and is defined by a power function.
(10)$${\mathrm{Reflection}(x,y)=[\mathrm{Reflection}(x,y)]}^{\frac{1}{\gamma }}\;\;\;\;\;\;\;\gamma \in \left[1,\;10\right]$$

#### 3.2.3. Frequency-Tuned Salient Region Extraction

In this paper, the MSR algorithm is improved by adapting the frequency-tuned (FT) saliency detection, which analyzes images from a frequency perspective and uses more information from the low-frequency targets with higher temperatures. Figure 4 shows the structure of multi-scale retinex algorithm based on frequency-tuned saliency detection (MSRFT). In the actual calculation, it uses the center-periphery operator of color features to obtain the saliency map.
(11)$$I(x,y)=∥I_{\mu }-I_{\mathrm{whc}}\left(x,y\right)∥$$
where $I_{\mu }$ is the arithmetic mean of image pixels, $I_{\mathrm{whc}}\left(x,y\right)$ is the original image after Gaussian blur with a window of 5×5, and ∥∗∥ is the Euclidean distance, which can be further revised as:(12)$$I(x,y)=∥I_{\mu }^{\mathrm{MSR}}\left(x,y\right)-I_{\mathrm{whc}}^{\mathrm{MSR}}\left(x,y\right)∥$$

### 3.3. Global Contrast Based Saliency Map Detection (LCHE)

#### 3.3.1. Global Contrast Based Salient Region Extraction

In environments where image pixels are low, target pedestrians and vehicles are too small to be distinguished from the background. Moreover, along with a lot of noise, the saliency detection algorithm can effectively detect and extract the region of interest (ROI) of the image. Under the condition of low resolution, it can effectively highlight the foreground objects and weaken the background information. It analyzes and processes the image through three salient features which are orientation, color, and illuminance, as shown in Figure 5. The feature map is generated through the center-periphery operator, and the three feature saliency maps are generated after merging and normalization. Finally, the final map is obtained by using linear weighting [34].

The global contrast-based salient region detection (LC), performs better than the RC algorithm (region-based contrast) and HC algorithm (histogram-based contrast). This algorithm is designed with a linear computational complexity concerning the number of image pixels. The saliency map of an image is built upon the color contrast between image pixels. The saliency value of a pixel in an image *I_k_* is defined as:(13)$${S(I}_{k})=\sum_{j-1}^{n}f_{j}∥I_{k}-I_{i}∥\;\;\;\;\;\;\;I_{i}\in \left[0,\;255\right]$$
(14)$$\mathrm{Final}(k)=\frac{{S(I}_{k})-S_{\min}}{S_{\max}-{\;S}_{\min}}\times 255$$
where the ∥∗∥ represents the grayscale distance metric, the $f_{j}$ represents the frequency of a certain color, and the number of all colors is *n*, *Final*(*k*) is the final gray value.

#### 3.3.2. Grayscale Distribution Equalization

As an important parameter of infrared images, the gray level can effectively distinguish the foreground object and background environment of the infrared image. The equalizing histogram (HE) adjusts the grayscale distribution of the image by increasing the grayscale and expanding the grayscale range concerning pixels with larger grayscale density. At the same time, it assigns fewer gray levels concerning pixels with lower grayscale density to achieve the purpose of enhancing the image contrast.
(15)$$H(k)=\sum_{j=0}^{k}H(j)\;\;\;\;\;(0\leq k\leq 255)$$
(16)$${\mathrm{Final}}^{\prime }(k)\;=\;\lfloor \frac{255\times H(k)}{P(255)}\rfloor$$
where *Final′*(*k*) stands for the new gray value of the pixel with gray level *k* after equalization, *H*(*k*) is the cumulative histogram of the image, and *H*(*j*) is the statistical histogram of the image.

In this paper, a global contrast-based salient region detection (LC) is used to highlight objects of interest in the foreground under prosthetic vision based on the HE algorithm, and then Gaussian Filter is used to decrease the noise of the saliency map.

### 3.4. Object Detector Based on Deep Learning

#### 3.4.1. Network Structural Improvement

As a one-stage network, (You Only Look Once) (YOLOv4) has many better properties compared with the previous versions of YOLO. In the backbone part, the CSPDarknet53 is selected as the backbone network to solve the problem of the large volume of calculations in reasoning from the perspective of network structure design. For the head part, this network applies Complete IoU loss (CIoU loss) to evaluate the performance of the detection, which makes the bounding box more accurate and has a high object recall. For data augmentation, the definition of Bag of Freebies in YOLOv4 (BoF) can ensure that the object detector achieves higher robustness to the images obtained from different environments. Moreover, in the YOLOv4 structure, the cross-stage-partial is added to Darknet-53 compared with YOLOv3, which can gain a higher accuracy as well as a high speed [35]. The cross-stage connection and the residual part can maintain these small-scale features, thus this paper did not need to change the connection relationship of the original YOLOv4 structure.

#### 3.4.2. Pre-Trained Models

In this paper, the FLIR dataset, which contains 8862 training images and 1366 testing images with 640 × 512 pixels, and pre-trained YOLOv4 weights via MSCOCO were used to train the initial model, model with modified active function, and model with both modified active function and network batch size. The performance of object detectors in inclement weather such as nighttime and rainy day is shown in Table 1. The initial parameters of the YOLOv4′s training stage were set as follows: the activate function was the Swish activation function; the training epoch was 120; the batch size was 64 using the Cosine Annealing Algorithm; the initial learning rate was 10 × 10^−3^; the max batch was 8862; the steps were 7090 and 7976, and the momentum was 0.9; Intersection over Union (IoU) threshold was 0.5.

As is shown in Figure 6, the YOLOv4 network rectified of the activating function and other parameters performed better over the other detectors such as YOLO, YOLOv2, YOLOv3, YOLOv3-tiny, and YOLOv4-tiny [22]. Our group chose the YOLOv4 network to detect the infrared images in this paper, which contain 215 nighttime images, 215 fog images, 209 rain images, and 625 snow images.

## 4. Experiments and Results

### 4.1. Low-Visibility Infrared Datasets

In this paper, to reflect the comparison of differences in infrared images in different harsh environments, the Low-visibility Infrared (LVIR) datasets were selected to be acquired according to different locations and different devices, thus illustrating the advantages and disadvantages of infrared image enhancement algorithms. Figure 7 shows some example images of LVIR datasets. Furthermore, our study provides a new orientation to conduct intelligent transportation research and comprehensive infrared image enhancement methods based on infrared features under different extreme weather conditions.

Low-visibility Infrared Datasets include:Datasets collected in heavy rain in this paper, a total of 209 infrared images of rainy environments;Open-source datasets proposed by Miethig et al. [6], a total of 625 infrared images of snowy environments are used in this paper due to the hardly accessible snowy scenes;Open-source datasets introduced from GUIDE SENSING INFRARED database, a total of 215 infrared images each for nighttime and foggy weather. All the detailed information of each dataset and their uses in this paper are detailed in Table 2.

### 4.2. Performances of Improved Enhancement Methods

To improve the detection accuracy of infrared targets in low-visibility conditions such as rainy days, the specific enhancement algorithms were readjusted to adapt to infrared features in different environments. After the improvements of three mainstream enhancement algorithms, infrared images of nighttime, rain, snow, and fog were enhanced effectively, which highlights the targets and dampens the backgrounds of these images.

Compared with the classic wavelet transform, original images after bilateral filtering show comparable performance in enhancement steps using wavelet transform. As is shown in Figure 8, pedestrians in the rain can be better enhanced after improvement of the bilateral filter.

Relatively, based on bilateral filtering, the retinex algorithm adjusts images with darker tones and balances the overall brightness distribution while enhancing the proportion of the region of interest in the image. Figure 9 shows the results of SSR, MSR, and MSRFT algorithms. The retinex algorithm combined with FT saliency detection can highlight the infrared target, and this improved algorithm is effective in foggy environments.

The saliency map detection (Figure 10) can effectively detect and extract the region of interest (ROI) of the image and highlight the foreground objects and weaken the background information. Among them, FT algorithm and LC algorithm both have better effects on infrared target enhancement at nighttime and in snow weather, while the HC algorithm performs otherwise. And the LC algorithm is utilized and improved in this paper.

Through experiments, first, the improved wavelet transforms (WTBF) performed well in rainy weather, where the high-frequency coefficients and background information which became wetted similar to umbrellas were almost weakened by threshold denoising to enhance the infrared targets such as pedestrians; second, the improved MSR algorithm (MSRFT) had a better performance for low-contrast and dim targets in foggy weather; third, the global contrast based salient region extraction (LCHE) could strongly enhance the infrared targets at nighttime and in snowy weather.

### 4.3. Evaluation Indicators

#### 4.3.1. Peak Signal-to-Noise Ratio

The peak signal-to-noise ratio (*PSNR*) is calculated in this paper to represent the ratio of the maximum possible power representing the signal to the destructive noise power that affects the accuracy of its representation, which further indicates the quality of the enhanced image compared with the original image. To calculate *PSNR*, the calculation of Mean Square Error (*MSE*) is needed. Two monochrome images *I* and *K*, if one is a noisy approximation of the other, then their *MSE* is defined as:(17)$$\mathrm{MSE}=\frac{1}{\mathrm{mn}}\sum_{i=0}^{m-1}\sum_{j=0}^{n-1}{\left[I(i,j\right)-K(i,j)]}^{2}$$
(18)$$\mathrm{RMSE}=\sqrt{\mathrm{MSE}}$$

The concept of *RMSE* is a root mean square error, and *PSNR* is obtained through *RMSE* in this paper:(19)$$\mathrm{PSNR}=20\times \log_{10}(\frac{\mathrm{PIXEL}\_\mathrm{MAX}}{\mathrm{RMSE}})$$

For monochrome or gray-level images, the *PIXEL_MAX* is 255.0. The evaluation rules of *PSNR* are shown in Table 3:

In this paper, the *PSNR* results of enhanced infrared images are calculated, and the plot in Figure 11 shows the results of *PSNR* values of each condition. The “Official” means the infrared raw images without any processing. Exceptionally rainy targets, a dash of *PSNR* values in nighttime, snow, and fog scenes appear abnormal due to low-clearance infrared objectives, and lack of edge information.

Based on the plots, firstly, after augmenting improved LCHE, the *PSNR* values of enhanced images in fog weather rarely increased due to the substantial increase in only a small range of grayscale value of thermal images which further brought about the overall decrease in adjacent pixel correlation when calculating *SSIM*. Secondly, the improved WTBF is predominant in augmenting infrared targets in rainy weather, corresponding to a mean *PSNR*(WTBF)/*PSNR*(WT) pair of 38.58 dB/22.20 dB. Finally, the *PSNR* values of foggy infrared images increase from 42.10 dB to 42.24 dB after being augmented by MSRFT.

#### 4.3.2. Structural Similarity Index

To measure the structural correlation between two images, the Structural Similarity Index (SSIM) defines structural information as independent of brightness and contrast from the perspective of image composition, reflecting the correlation between adjacent pixels. Moreover, the correlation expresses the structural information of objects in the scene which is calculated among the enhanced images. Distortion is modeled as a combination of three different factors, brightness, contrast, and structure. The variables of formula (20) are explained in Table 4.
(20)$$\mathrm{SSIM}(x,y)=\frac{\left(2\mu_{x}\mu_{y}+c_{1})(\sigma_{\mathrm{xy}}+c_{2}\right)}{\left(\mu_{x}^{2}+\mu_{y}^{2}+c_{1})(\sigma_{x}^{2}+\sigma_{y}^{2}+c_{2}\right)}$$

As is indicated above in Figure 12, the *SSIM* value of enhanced images mostly arises, which interprets the validity of enhanced images in structural similarity compared with original images. Table 5 shows the mean *SSIM* value of fog condition dropped by 0.03, there are compacter values of *SSIM* unveiled by a lower standard deviation of 0.024.

#### 4.3.3. Target Detection Performance

To verify that the enhanced image can obtain higher detection accuracy, the article used the detector to detect the original infrared image and the enhanced image separately. The number of randomly selected detection images in different scenes is 100 each, which means 100 original images and 100 enhanced images each, for a total of 800 images in four scenes, and the detection categories are mainly cars and people. 

Table 6 shows the accuracy results of detecting the original IR image and its enhanced image with the same detector separately, where the main categories are “people” and “car”. The “Official” is the infrared raw images without any processing. Compared with the detection of official infrared images, the detection accuracy of “car” and “people” on rainy days increases by 12% and 1%, respectively. Since the “people” category is already brighter than the backgrounds of “Official” images, the improved accuracy is smaller. 

In the foggy environment, the results show that the enhanced image is unable to detect “people” and “car”. One of the main reasons is that MSRFT enhances the building lights, streetlights, and even white lane lines in the background along with the enhancement. So that the enhanced images have overlapping lights with “people” and “car”, the detector trained using the FLIR dataset cannot accurately distinguish the targets of enhanced fog images. 

In the night environment, “car” detection accuracy improves by 1.2%, while the night images are mainly of expressways, so the “people” category is missing. The night image enhanced by LCHE has more room for improvement, for example, in the detection of long-range targets.

Finally, in the snowy environment, due to the low resolution of the Official image itself, the foreground target brightness is enhanced and the background is darkened by the algorithm enhancement, improving the final detection accuracy by 3% and reaching 90% accuracy. However, there is a decrease in accuracy when detecting “people”, considering that it may be because most of the external clothing temperature of pedestrians blends with the environment, and the enhancement only enhances the part of the bright temperature area, which cannot be accurately judged during “people” target detection. This slight decrease in accuracy for “people” detection requires more research. 

When performing target detection, both the size and the resolution of the input image have an impact on the final detection result. In general, the detection effect of infrared images after enhancement is not as high as that achieved by visible images, such as up to 99% accuracy in the field of semantic segmentation, which also indicates that there is more room for improvement in infrared target detection.

## 5. Conclusions

The results show that the temperature information and pixel distribution information of infrared images are the key factors to be considered in image enhancement algorithms. For objective enhancement, the MSRFT methods achieve the highest *PSNR* value of 42.24 dB in fog weather and the WTBF methods appear steep to increase *PSNR* by 16.38 dB of rainy targets. *SSIM* indicates the excellence of improved enhancement methods overall. The *SSIM* values are 0.96, 0.61, 0.81, and 0.90 in rain, fog, nighttime, and snow, respectively. After detection of the target by the YOLOv4 deep learning detector, it can be concluded that the enhanced IR images generally improve in target detection accuracy. In rainy weather, the detection accuracy of cars and people on rainy days both increased. At nighttime, enhanced images with some improvement in car detection reach 85.6%. In snowy weather, the detection accuracy of cars in snowy environments is improved by 3%, which achieves better results in the image detection field.

Moreover, the proposed WTBF enhancement method needs less time than the other three methods and saves more computer memory usage. For object detection, compared with the Official images, the enhanced infrared images save about 20% of detection time with an accuracy improvement of rain, nighttime, and snow images.

In conclusion, the enhancement algorithms designed according to the image characteristics can effectively improve the accuracy of the target detection, which demonstrates the high accuracy, adaptability, and efficiency of the proposed methods. Future research directions of this paper may include the saliency region reflected by the target temperature difference and enhancing the color representation of this region so that the target is salient.

## Figures and Tables

**Figure 1 sensors-22-05835-f001:**
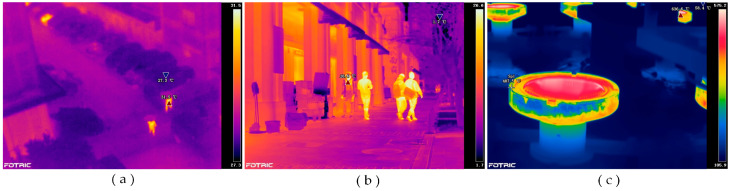
Normal samples of thermal imaging in different applications. (**a**) IR surveillance in hurricane weather; (**b**) detect pedestrians at nighttime; (**c**) quality inspection of iron and steel parts.

**Figure 2 sensors-22-05835-f002:**
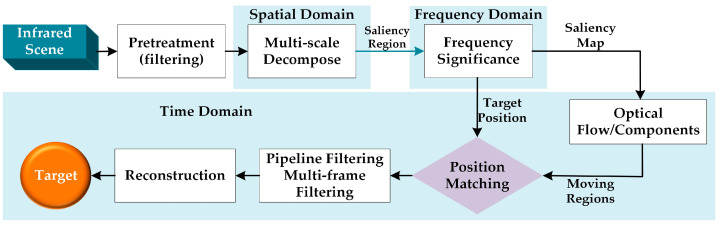
Diagram of the proposed image enhancement algorithm.

**Figure 3 sensors-22-05835-f003:**
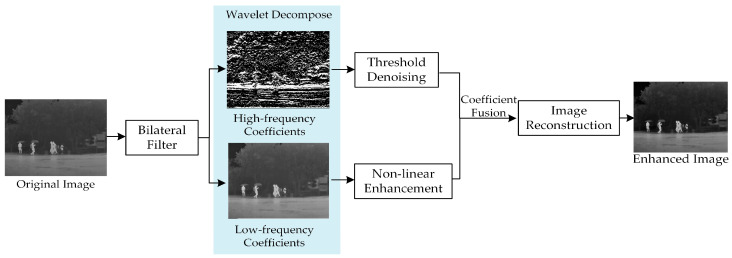
Structure of wavelet transformation algorithm based on a bilateral filter.

**Figure 4 sensors-22-05835-f004:**
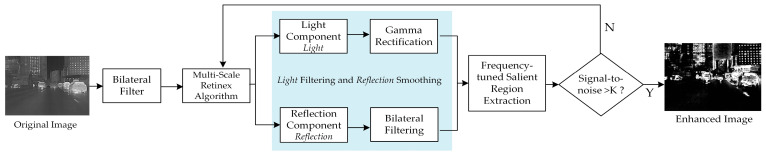
Structure of multi-scale retinex algorithm based on frequency-tuned saliency detection (MSRFT).

**Figure 5 sensors-22-05835-f005:**
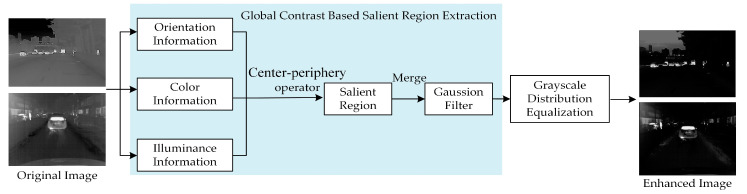
Improved saliency detection based on HE.

**Figure 6 sensors-22-05835-f006:**
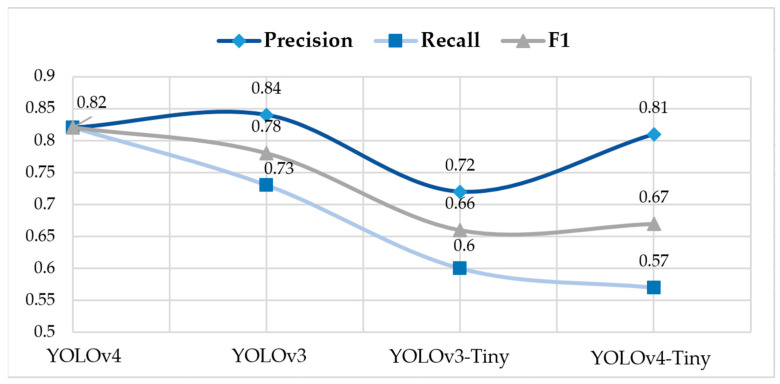
The graph of trends of Precision, Recall, and F1 per version of YOLO.

**Figure 7 sensors-22-05835-f007:**
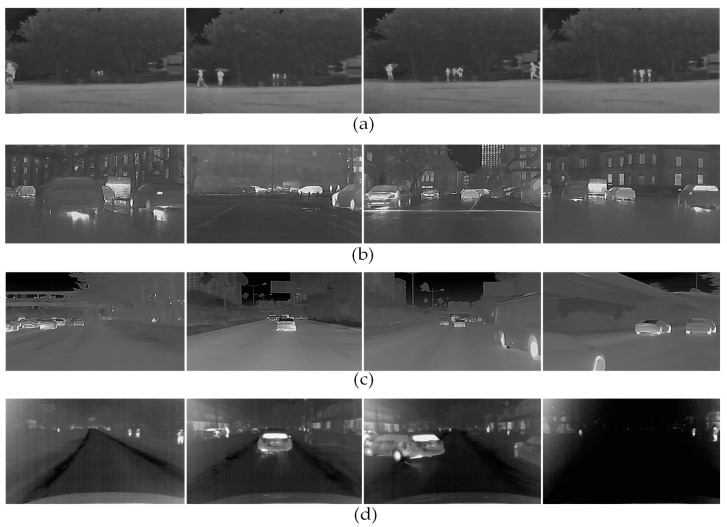
Low-Visibility Infrared (LVIR) datasets. (**a**) IR images in rainy weather; (**b**) IR images in foggy weather; (**c**) IR images at nighttime; (**d**) IR images in snowy weather.

**Figure 8 sensors-22-05835-f008:**
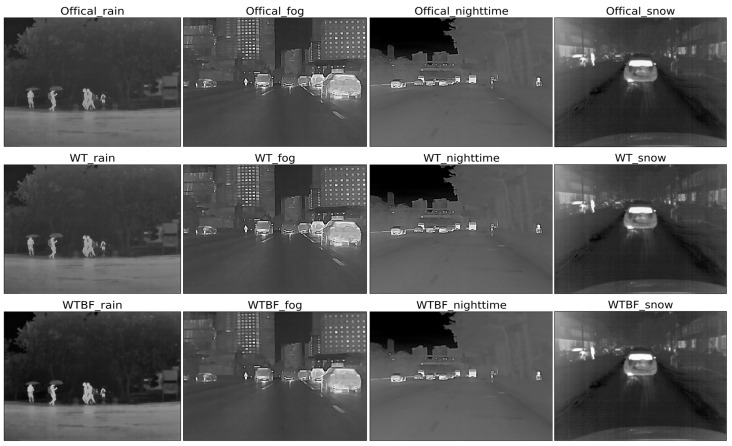
The enhancement results of WT and WTBF. The first row is the original images of the four scenes (rain, fog, night, and snow), the second row is the results of the WT method, and the third row is the results of the WTBF method.

**Figure 9 sensors-22-05835-f009:**
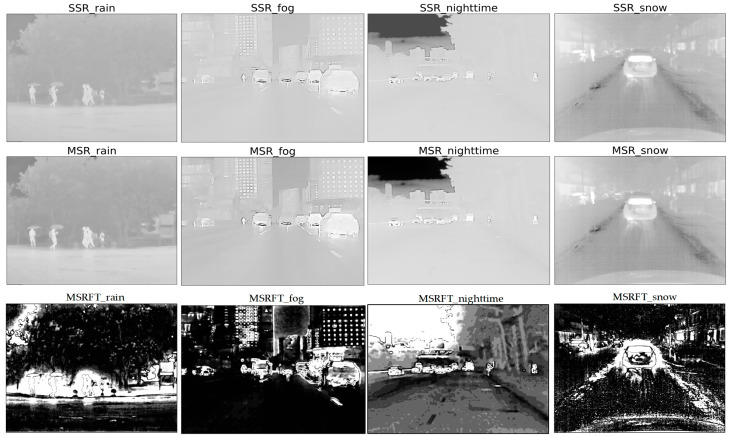
The enhancement results of SSR, MSR, and MSRFT. The first row is the results of the SSR method of the four scenes (rain, fog, night, and snow), the second row is the results of the MSR method, and the third row is the results of the MSRFT method.

**Figure 10 sensors-22-05835-f010:**
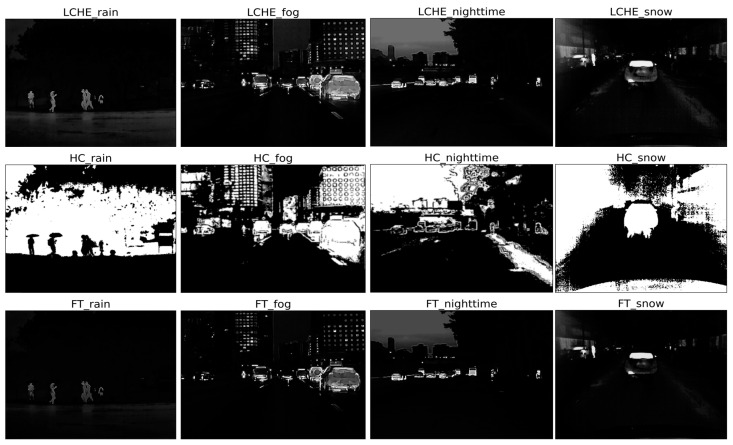
The enhancement results of LCHE, HC, and FT. The first row is the results of the LCHE method of the four scenes (rain, fog, night, and snow), the second row is the results of the HC method, and the third row is the results of the FT method.

**Figure 11 sensors-22-05835-f011:**
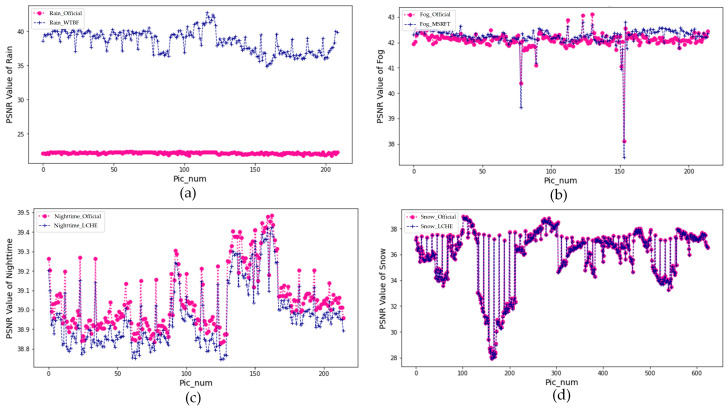
The scatter plot of *PSNR* values in four scenes, (**a**) *PSNR* values of rain images; (**b**) *PSNR* values of fog images; (**c**) *PSNR* values of nighttime images; (**d**) *PSNR* values of snow images. WTBF is applied in rainy scenes, MSRFT detection is applied in the foggy scene, and LCHE is applied in nighttime and snowy scenes.

**Figure 12 sensors-22-05835-f012:**
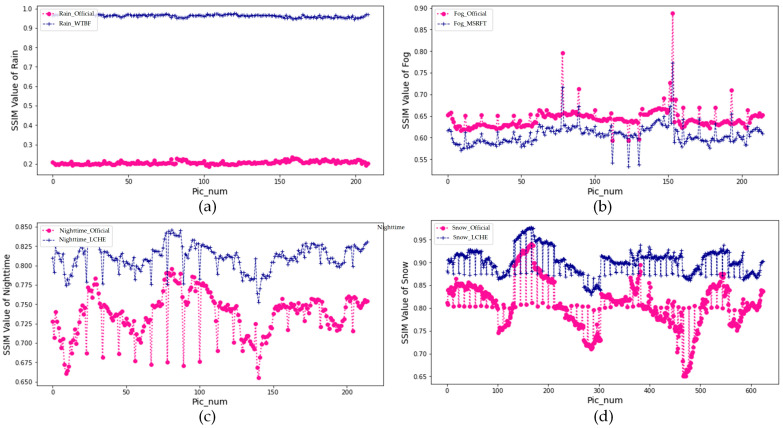
The scatter plot of *SSIM* values in four scenes, (**a**) *SSIM* values of rain images; (**b**) *SSIM* values of fog images; (**c**) *SSIM* values of nighttime images; (**d**) *SSIM* values of snow images. WTBF is applied in rainy scenes, MSRFT detection is applied in the foggy scene, and LCHE is applied in nighttime and snowy scenes.

**Table 1 sensors-22-05835-t001:** Performance of different YOLO network detectors trained before experiments.

Version	Backbone	Precision	Recall	F1
YOLOv4	CSPDarknet53	0.82	0.82	0.82
YOLOv3	Darknet-53	0.84	0.73	0.78
YOLOv3-Tiny	Darknet-53	0.72	0.60	0.66
YOLOv4-Tiny	CSPDarknet53	0.81	0.57	0.67

**Table 2 sensors-22-05835-t002:** Information on the LVIR datasets and their uses in enhancement and detection phases.

Data Source	Ambient Temperature	Weather Condition	Resolution	Enhancement Phase	Detection Phase
This paper	15 °C	Heavy rain	640 × 512	209	100
Open source	13 °C	Heavy fog	384 × 288	215	100
Open source	17 °C	Nighttime	384 × 288	215	100
Open source	0 °C	Heavy snow	875 × 700	625	100

**Table 3 sensors-22-05835-t003:** The evaluation rules of *PSNR* value.

** *PSNR* **	**Value (dB)**	**Evaluation**
	≥40	Excellent
	30–40	Good
	20–30	Worse
	≤20	Unacceptable

**Table 4 sensors-22-05835-t004:** The definition of variables of the above *SSIM* formula.

Variables	Definition
$\mu_{x}$	The mean of *x*
$\mu_{y}$	The mean of *y*
$\sigma_{x}^{2}$	The variance of *x*
$\sigma_{y}^{2}$	The variance of *y*
$\sigma_{xy}$	The covariance of *x* and *y*
$c_{1}$, $c_{2}$	$c_{1}$ $={\left(k_{1}L\right)}^{2},c_{2}$ = (*k*_2_*L*)^2^
*L*	The scale of the pixel, *L* = 255.0
*k*_1_, *k*_2_	*k*_1_ = 0.01, *k*_2_ = 0.03

**Table 5 sensors-22-05835-t005:** The mean *SSIM* values and standard deviation.

Low-Visibility Scenes	*SSIM*	
	Mean	Standard Deviation
Rain_Official	0.21	0.008
Rain_WTBF	0.96	0.007
Fog_Official	0.64	0.026
Fog_MSRFT	0.61	0.024
Nighttime_Official	0.74	0.029
Nighttime_LCHE	0.81	0.017
Snow_Official	0.81	0.053
Snow_LCHE	0.90	0.029

**Table 6 sensors-22-05835-t006:** The accuracy in detection of “car” and “people” categories under four low-visibility scenes.

Low-Visibility Scenes	Object Detector (Backbone)	Accuracy	
		Car	People
Rain_Official	YOLOv4 Network (CSPDarknet53)	0.360	0.850
Rain_WTBF		0.480	0.860
Fog_Official		0.87	0.46
Fog_MSRFT		/	/
Nighttime_Official		0.844	/
Nighttime_LCHE		0.856	/
Snow_Official		0.870	0.760
Snow_LCHE		0.90	0.740

## Data Availability

No new data were created or analyzed in this study. Data sharing is not applicable to this article.

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
