# Peer review of "Infrared Sensation-Based Salient Targets Enhancement Methods in Low-Visibility Scenes"

_sensors, 2022, doi:10.3390/s22155835_

Round 1

Reviewer 1 Report

The author proposed three infrared image enhancement methods, a threshold denoising model for rainy environments based on wavelet transform and bilateral filtering (WTBF), and a salient component enhancement method based on multi-scale retinex algorithm combined with frequency-tuned salient region extraction (MSRFT), and using salient features to improve image contrast and noise distribution methods. Finally, the accuracy of the bounding box of enhanced images was tested by the pre-trained and improved object detector. Then they use the target detection algorithm to verify the effectiveness of the enhancement algorithm. The following issues need to be addressed:

1. Can a more comprehensive AP value be used to measure the effectiveness of the infrared image enhancement algorithm using target detection, such as Table 6, can it be improved?

2. Is there a simple explanation for the decrease in the SSIM value under fog conditions, and does it have some impact on the performance of target detection?

Author Response

Dear Reviewer 1:

Thank you very much for reviewing our manuscript entitled “Infrared Sensation-based Salient Targets Enhancement Methods in Low-visibility Scenes”(ID: 1834450). And on behalf of my co-authors, we thank you very much for giving us an opportunity to revise our manuscript.

The comments and suggestions are all valuable and helpful for revising and improving our paper. The main corrections in the paper and the responses to your comments can be seen in the attachment file. Please see the attchment.

Reviewer 2 Report

The paper is of interest, anyway, it suffers in three key points, i.e.: readability, knowledge of the published scientific literature, and appropriate discussion of the results obtained.

^ To the abstract: “…and due to the poor image quality, …”. This is the personal thinking of the authors; in this case, the term “poor” is not true because it depends on many factors.

^ Authors should refrain from using personal pronouns such as "we" throughout the text and I encourage them to write it in an impersonal form of writing.

^ To the keywords: I suggest adding “wavelet transform”

^ The first phrase of the Introduction section is for sure of great impact but it lacks a scientific point of view. Please, re-phrase or add a Ref.

^ To the Introduction: 8~15um wavelength -> 8~15μm wavelength

^ To the Introduction: “Meanwhile, IR technology is widely used in military and civilian applications”. This phrase is missing important sectors which must be taken into account, as well as appropriate references must be added. Please re-organize it as in the following: “Meanwhile, IR technology is widely used in military, cultural heritage, industrial, medical, biological and civilian applications” [a-f].

[a] DOI: 10.1080/17686733.2020.1793284

[b] DOI: 10.1080/17686733.2020.1799304

[c] DOI: 10.1080/17686733.2020.1846113

[d] DOI: 10.1080/17686733.2020.1793283

[e] DOI: 10.1080/17686733.2021.1882075

[f] DOI: 10.1080/17686733.2020.1805939

^ To Fig. 1: Please add the false color temperature scale in each sub-figure (a-c).

^ In the brief recap proposed by the authors in section 2 to summarize the scientific literature, some Refs. seem currently missing. I refer to:

DOI: 10.1080/17686733.2020.1786640

DOI: 10.1080/17686733.2019.1586376

DOI: 10.1080/17686733.2018.1557453

Please add them to the revised version of the paper.

^ I suggest adding, in addition to Tab. 1, a linked graph in order to show clearly to the reader the trends of Precision, Recall, F1 per Version of YOLO.

^ The results summarized in Tab. 6 should be discussed more in-depth.

^ Please add in the Conclusions section a phrase regarding the computational cost of the proposed algorithm with respect to the ones discussed in the paper.

In synthesis: major revisions are required before the final acceptance for publication.

Author Response

Dear Reviewer 2:

Thank you very much for reviewing our manuscript entitled “Infrared Sensation-based Salient Targets Enhancement Methods in Low-visibility Scenes”(ID: 1834450). And on behalf of my co-authors, we thank you very much for giving us an opportunity to revise our manuscript.

The comments and suggestions are all valuable and helpful for revising and improving our paper. The main corrections in the paper and the responses to your comments can be seen in the attachment file. Please see the attchment.

Round 2

Reviewer 2 Report

Dear Authors: great job during the revision process! The paper is accepted; anyway, I would like suggesting to check Ref. [7] during the proof correction. Indeed, it is currently missing some important data, i.e.: name of the Journal, year of publication; volume/issue, pages, and DOI. Please add them later.